# Effects of Land Cover on the Taxonomic and Functional Diversity of the Bird Communities on an Urban Subtropical Mountain

Wenwen Zhang [1,†], Shengjun Zhao [1,†], Xiao Yang [1], Jing Tian [1], Xue Wang [2,*], Ding Chen [3], Yuan Yu [4], Jie Shi [1], Peng Cui [1,*] and Chunlin Li [5]

1   State Environmental Protection Key Laboratory on Biodiversity and Biosafety, Nanjing Institute of Environmental Sciences, Ministry of Ecology and Environment of the People's Republic of China, Nanjing 210042, China; zhangwenwen@nies.org (W.Z.); zhaoshengjun@nies.org (S.Z.); yx2425471650@163.com (X.Y.); tianjingjing444@163.com (J.T.); shijie.ahu@gmail.com (J.S.)
2   Jiangsu Academy of Forestry, Nanjing 211153, China
3   Zhejiang Environmental Protection Group, Hangzhou 310012, China; 15657218780@163.com
4   Academy of Teaching and Education, Monash University, Wellington Rd, Clayton, VIC 3800, Australia; yuyuan2024sq@outlook.com
5   School of Resources and Environmental Engineering, Anhui University, Hefei 230601, China; lichunlin1985@163.com
*   Correspondence: xuewls@163.com (X.W.); cuipeng@nies.org (P.C.)
†   These authors contributed equally to this work.

**Abstract:** Mountain ecosystems are crucial for global biodiversity conservation. However, their landscape features are constantly changing owing to urban expansion. Understanding the relationships between biotic communities and landscape features is essential for biodiversity conservation. This study aimed to examine the effect of land cover type on avian communities in Lishui, a mountainous urban area in eastern China. Avian surveys were conducted using 168 line transects in total across different land cover types once per season from December 2019 to January 2021. We assessed the diversity of bird communities by calculating various metrics at both taxonomic and functional levels. Among the land cover types measured, woodland, built-up land, cultivated land, and water bodies significantly influenced bird community diversity and composition. Species richness, species abundance, and functional richness were negatively correlated with the proportion of woodland but were positively correlated with the proportion of non-natural land cover, such as built-up and cultivated land. In contrast, functional evenness was positively correlated with the proportion of woodland and grassland but negatively correlated with the proportion of non-natural land cover. Land cover type also exhibited significant correlations with avian functional characteristics such as diet, foraging strata, and body mass, thereby influencing the overall community structure. Our results indicated that mountainous landscape patterns substantially affect avian communities. Different land cover types possess varying resource endowments that affect the distribution of avian species. Therefore, urban landscape planning in mountainous areas should carefully consider the various functions provided to organisms by different types of land cover to promote biodiversity.

**Keywords:** bird community; functional diversity; taxonomic diversity; landscape pattern; mountain city

## 1. Introduction

Mountain ecosystems are an important component of terrestrial ecosystems worldwide and are considered pivotal areas for biodiversity conservation [1]. Although mountains comprise only 25% of the Earth's land area, they are home to approximately 85% of the world's wild animals [2,3]. However, human population growth and economic development have accelerated deforestation toward clear land for agricultural or residential

needs [4]. These activities have intensified the modification of natural landscapes in mountainous regions, imposing considerable pressure on the species inhabiting these areas [1].

In recent years, extensive research has been conducted investigating the impact of natural landscape modifications on biodiversity. In particular, birds, which occupy a diverse range of habitats and are highly susceptible to environmental fluctuations [5], have become the most frequently studied species for examining the effects of anthropogenic habitat alterations on animals [6–8].

Anthropic landscape changes encompassing alterations in land cover type, quantity, and composition often result in a loss of or increase in avian species and changes in community species composition [9–11]. On the one hand, landscapes that encompass a substantial proportion of natural or semi-natural land cover tend to have positive effects on species richness and abundance. This may be attributed to the provision of ample natural shelter and high habitat connectivity in such landscapes [12–14]. However, anthropogenic landscape modification often reduces natural habitats, leading to a fragmentation of the remaining native land cover into isolated patches within a matrix of non-natural land-cover types [10,14]. According to the habitat heterogeneity hypothesis, heterogeneous habitats generally contain more species because they provide more diverse ways of exploiting the available environmental resources (niches) [15]. However, access to resources may be impeded or entirely prevented when isolated native patches are encompassed by non-natural land cover that is unsuitable for survival [16]. On the other hand, anthropogenic landscapes can also provide benefits to certain species in some cases [17,18]. For instance, some habitat generalists, such as house sparrows and barn swallows, are found in diverse arrays of anthropogenic habitats [19], whereas other species are endemic to a particular type of natural habitat [20]. Therefore, understanding the potential responses of different species to various forms of anthropogenic land cover change is crucial to effectively guiding urban landscape planning in mountainous regions.

The ecological traits of a species influence its ability to thrive in specific environments [21]. Environmental factors act as selective forces that eliminate species that cannot tolerate the conditions at a particular site [22]. Hence, in addition to taxonomic modifications (such as variations in species richness, abundance, and composition), biomes also exhibit functional changes owing to alterations in landscape patterns [5,23]. However, numerous studies have demonstrated that the pattern of changes in functional diversity may differ from that of taxonomic diversity [6,24]. For instance, the study conducted by Coetzee and Chown (2016) found that land use change resulted in an increase in the diversity of avian species in the local area [6]. However, land use change also caused a decrease in the abundance of species that possess unique functional traits. The simultaneous consideration of taxonomic and functional diversity has the potential to enhance our understanding of the impact of anthropogenic habitat alteration on biological communities.

China's terrain is predominantly mountainous, with approximately two-thirds of its land area covered by mountains. Furthermore, approximately one-third of cities in China are located in mountainous areas [25]. The expansion of urban areas is an unavoidable consequence of the exponential development of China's economy. In this study, we examined the impact of landscape patterns on bird communities in a mountainous city in eastern China. Specifically, we conducted a comprehensive analysis of bird community diversity by calculating multiple metrics at both the taxonomic and functional levels, and assessed the impact of landscape patterns on these metrics. Furthermore, we conducted an analysis to investigate the relationship between species composition and the functional traits of assemblage and landscape patterns. We hypothesized that (1) the diversity of bird species, taxonomically and functionally, would increase to some extent owing to the emergence of non-natural land cover and expansion of bioavailable niches; (2) different land cover types may have varying associations with the functional characteristics of birds, thereby influencing the composition of bird communities; and (3) the community structure and diversity of birds may vary between seasons.

## 2. Materials and Methods

### 2.1. Study Region

Lishui city, located in the Zhejiang province of eastern China (27°25′–28°57′ N; 118°41′–120°26′ E), is a typical mountainous urban area. It spans a total area of 17,275 km² and is home to a population of 2.70 million residents (Figure 1). Mountains account for more than 90% of the total land area. Within this region, more than 3500 peaks exceed an elevation of 1000 m, with 240 peaks surpassing 1500 m. The region has a subtropical monsoon climate and is characterized by a wide range of vegetation types and substantial vegetation coverage, with forest cover exceeding 80%. Intricate and diverse terrains and landforms, along with a wide array of vegetation types, have resulted in the formation of numerous high-quality habitats that support a diverse range of invertebrate and vertebrate species [26].

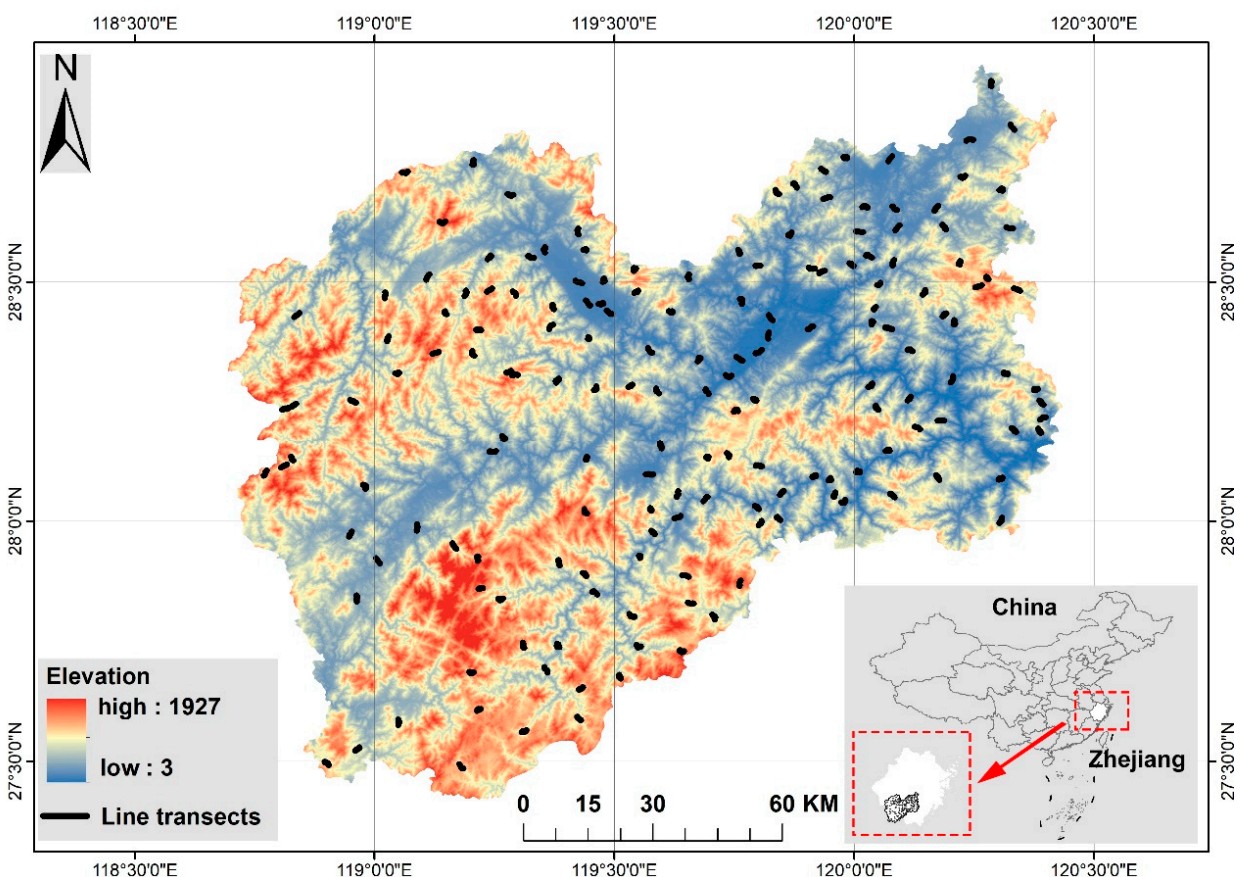

**Figure 1.** Location of study area and distribution of avian sampling line transects.

### 2.2. Bird Survey

We surveyed birds using line transects [27]. In total, 168 line transects, each measuring 1.5 km in length, were evenly distributed throughout the study region (Figure 1). Each transect was surveyed 4 times, once every season from December 2019 to November 2021 (Spring: April–May; Summer: July–August; Autumn: October–November; Winter: December–January). To mitigate the potential impact of light and temperature during different seasons, surveys were conducted within the specific timeframe of three hours after sunrise and three hours before sunset. Furthermore, the surveys were conducted only when weather conditions were optimal, specifically, when there was no rain or wind. All surveys were conducted by the same six well-trained observers along all transects and across four seasons. During the surveys, the observers conducted transect walks at a consistent speed of approximately 2.0 km/h. On average, 45 min was spent on each transect. They utilized binoculars to document all visible bird species and listened for

bird vocalizations to record the presence of non-visible species in the area. To prevent the duplication of records, birds that flew over the transect were not recorded. Additionally, as our objective was to investigate the impact of land cover on bird communities, rather than to measure the exact bird population densities, direct bird count data were used in the subsequent analyses without adjusting for detectability. The bird taxonomy and nomenclature used in this study adhered to the guidelines outlined in A Checklist on the Classification and Distribution of the Birds of China [28].

### 2.3. Land Cover Types and Diversity

We quantified the composition of land cover types around each sampling transect to determine the degree of human activity. First, Landsat 8 satellite images with a spatial resolution of $30 \times 30$ m were obtained from the geospatial data cloud (http://www.gscloud.cn/, accessed on 28 August 2020). We used a series of image preprocessing techniques, including image fusion, image mosaicing, radiometric calibration, and atmospheric correction to mitigate the influence of different error sources. Second, supervised classification was performed on the Landsat 8 satellite images using the maximum likelihood classifier in ENVI 5.3. Subsequently, the percentage of various land-cover types within a buffer area (750 m radius) surrounding each sampling transect was computed using Geographic Information System processing tools. Ultimately, seven land cover types (woodland, cultivated land, shrubland, bare land, grassland, water bodies, and built-up land) were identified and further verified during field surveys. Additionally, we calculated the Shannon–Wiener diversity index of land cover types to characterize the landscape diversity within each sampling transect.

### 2.4. Functional Traits

To assess functional diversity, we selected 12 avian functional traits associated with resource utilization (Table 1). These traits included body mass (a continuous trait), diet (five categorical traits), and foraging strata (six continuous traits). Functional traits were derived from a global dataset provided by Wilman et al. (2014) [29], in which each species is assigned to a specific dietary guild, and the foraging strata for each species are represented as percentages in multiple columns that sum up to 100.

**Table 1.** Traits used for the estimation of functional diversity.

| Trait Type | Traits | Categories |
|---|---|---|
| Resource quantity | Body mass | Continuous |
| Diet guild | PlantSeed (feeding on plant and seeds);<br>FruiNect (feeding on fruits and nectar);<br>Invertebrate (feeding on invertebrates);<br>VertFishScav (feeding on vertebrates, fish and carrion);<br>Omnivore | categorical |
| Foraging stratum | Ground;<br>Understory;<br>Midstorey;<br>Canopy;<br>Air;<br>Water | Continuous |

### 2.5. Taxonomic and Functional Diversity

Taxonomic diversity was assessed by calculating species richness, species abundance, and the Shannon–Wiener diversity index for each line transect. Functional diversity was assessed by examining functional richness (FRic), functional evenness (FEve), and functional divergence (FDiv), each of which reveals distinct facets of functional diversity [30]. The FRic indicates the volume occupied by a species assemblage within a functional space, FEve reflects the regularity of the abundance distribution in the functional space, and FDiv is the degree to which the abundance of a community is distributed towards the extremities of the occupied trait space.

*2.6. Statistical Analyses*

We determined that the seven land cover types (percentage of area) were correlated and exhibited high collinearity; thus, we first performed principal component analysis to summarize these land cover types into three independent principal components (PC1, PC2, and PC3) that could explain most of the original variance.

To assess the effects of land cover type on the taxonomic and functional diversity of the avian community, we applied generalized linear models with Poisson distributions, and over-dispersion was adjusted using a quasi-Poisson procedure. In the models, PC1, PC2, PC3, LD, and season were used as explanatory variables. The associations between the three extracted components and the original explanatory continuous variables are presented in Table 2. In addition, we used the Wilcoxon rank sum test with Bonferroni correction to further examine the seasonal variability of avian diversity indices.

**Table 2.** Results of the principal components analysis of land cover types (percentage of area); PC = principal components.

| Land-Cover Types | PC1 | PC2 | PC3 |
|---|---|---|---|
| Woodland | 0.638 | 0.059 | 0.007 |
| Shrubland | 0.028 | 0.298 | −0.300 |
| Built-up area | −0.486 | 0.051 | −0.259 |
| Grassland | −0.127 | −0.099 | 0.848 |
| Water bodies | −0.281 | 0.573 | −0.069 |
| Cultivated land | −0.491 | −0.437 | −0.032 |
| Bare land | −0.143 | 0.614 | 0.345 |
| Proportion of variance (%) | 0.343 | 0.192 | 0.153 |
| Cumulative proportion (%) | 0.343 | 0.534 | 0.687 |

To assess the influence of land cover type on avian community composition, land cover data were integrated into the non-metric multidimensional scaling (NMDS) ordination space of avian community composition using the 'envfit' function in the 'vegan' R package. To improve the clarity of our findings, we visually represented the distribution of the avian dietary guilds in the ordination space. In this study, NMDS was calculated using a Bray–Curtis distance matrix based on species abundance data. Pearson's correlation analysis was used to evaluate the correlations between NMDS axis scores and each land cover type, and a permutation test (999 repetitions) was used to assess the significance of the correlation coefficients.

To further investigate the role of land cover type in shaping bird communities, we related the functional traits of bird species to land cover type using fourth-corner statistics [31]. By utilizing three datasets (i.e., species abundance, functional traits, and land-cover types), this methodology enabled the detection of positive or negative correlations between avian traits and the environmental characteristics of their respective habitats. Additionally, it provided a measure of the statistical significance of these associations.

All statistical analyses were performed using R version 4.0.1 [32], and $p < 0.05$ was considered significant.

## 3. Results

*3.1. Principal Component Analysis Results*

The first three principal components (PC1, PC2, and PC3) explained 69% of the land cover types (percentage of area) variation in the 168 samples (Table 2). In particular, PC1 (34.3%) was positively associated with woodland and negatively associated with built-up and cultivated land, PC2 (19.2%) was positively associated with both shrubland and water bodies, and PC3 (15.3%) was positively associated with grassland and negatively associated with shrubland.

### 3.2. Bird Survey Results

During the survey period, 45,027 birds from 316 species belonging to 18 orders and 67 families were recorded along the 168 line transects (Table S1). Among the recorded species, 59 were listed as Key Protected Wild Animal Species in China (38 in spring, 30 in summer, 44 in autumn, and 34 in winter), and 48 were listed as either near threatened, vulnerable, or endangered on the Red List of the International Union for Conservation of Nature (29 in spring, 19 in summer, 35 in autumn, and 31 in winter). The national key protected species accounted for 4.55% of the total number of individuals, while red-listed bird species accounted for 3.58%.

### 3.3. Influence of Land Cover Types on Avian Community Diversity

The generalized linear models showed that land cover types had significant impacts on the diversity of bird communities (Table 3). Specifically, PC1 (positively correlated with woodland but negatively correlated with built-up and cultivated land) was negatively correlated with species richness, species abundance, the Shannon–Wiener diversity index, and FRic, and positively correlated with FEve. Additionally, PC3 (positively correlated with grassland and negatively correlated with shrubland) was positively correlated with FEve.

**Table 3.** The effect of land cover types on bird diversity indices. Generalized linear model estimates of slopes of functions and their standard errors (in brackets) are presented. *** means $p < 0.001$; ** means $p < 0.01$; and * means $p < 0.05$. See Table 1 for an explanation of the principal components used in these analyses.

| Explanatory Variables | Models for | | | | | |
|---|---|---|---|---|---|---|
| | Richness | Abundance | Shannon | FRic | FEve | FDiv |
| PC1 | −0.08 (0.08) *** | −0.16 (0.02) *** | −0.05 (0.01) *** | −0.05 (0.10) *** | 0.01 (0.00) * | 0.00 (0.00) |
| PC2 | 0.03 (0.02) | 0.05 (0.04) | 0.01 (0.01) | 0.02 (0.01) | 0.00 (0.01) | 0.00 (0.01) |
| PC3 | 0.04 (0.02) | 0.03 (0.03) | 0.01 (0.02) | 0.01 (0.01) | −0.02 (0.01) * | 0.00 (0.02) |
| LD | 0.01 (0.01) | 0.05 (0.05) | −0.00 (0.01) | 0.01 (0.01) | −0.00 (0.00) | −0.00 (0.01) |
| Spring | −0.01 (0.06) | −0.34 (0.11) ** | 0.05 (0.02) * | 0.07 (0.04) | 0.02 (0.02) | −0.05 (0.01) *** |
| Summer | −0.12 (0.06) * | −0.48 (0.11) *** | 0.02 (0.02) | 0.14 (0.04) ** | 0.06 (0.02) *** | −0.09 (0.01) *** |
| Winter | −0.38 (0.07) *** | −0.45 (0.11) *** | −0.14 (0.02) *** | −0.22 (0.05) *** | −0.03 (0.02) | −0.02 (0.01) |

### 3.4. Influence of Season on Avian Community Diversity

The results from the generalized linear models also indicated that season was an important factor affecting the diversity indices, both at the taxonomic and functional levels (Table 3). The results of the Wilcoxon rank sum test showed (Figure 2) that the species richness was significantly higher in autumn (16.63 ± 11.49), spring (16.46 ± 9.44), and summer (14.70 ± 8.12) than in winter (11.27 ± 6.74). Additionally, the species abundance was significantly higher in autumn (90.24 ± 104.57) and spring (64.40 ± 56.65) than it was in winter (57.40 ± 92.73) and summer (55.97 ± 53.69). The Shannon–Wiener diversity index was the highest in spring (2.35 ± 0.53) and the lowest in winter (1.95 ± 0.45). It was also significantly higher in summer (2.28 ± 0.48) and autumn (2.24 ± 0.52) than it was in winter.

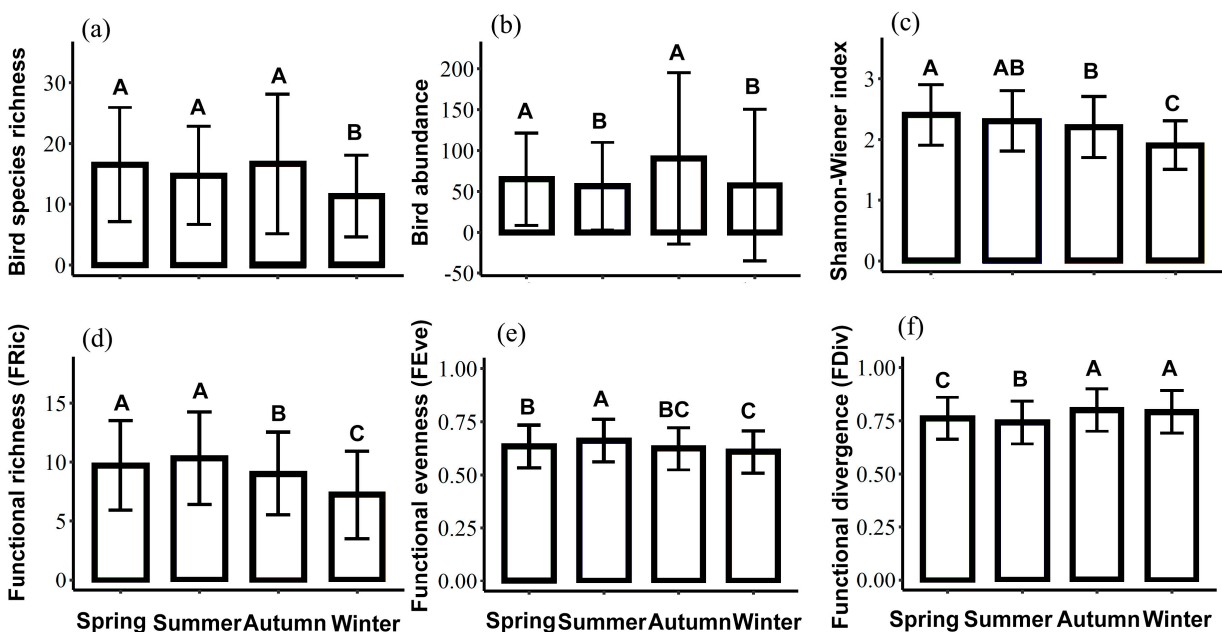

**Figure 2.** Metrics of the avian communities in different seasons: (**a**) species richness; (**b**) abundance; (**c**) Shannon–Wiener diversity index; (**d**) functional richness; (**e**) functional evenness; (**f**) functional divergence. The same letters indicate no significant differences between seasons, and significance was determined via the Wilcoxon rank sum test. The significance is $p \leq 0.05$.

In terms of functional diversity, FRic was the highest in summer (10.34 ± 3.92) and spring (9.65 ± 3.77), followed by autumn (8.98 ± 3.51) and winter (7.18 ± 3.69). FEve was the highest in summer (0.66 ± 0.09), followed by spring (0.63 ± 0.09) and autumn (0.62 ± 0.09), and it was the lowest in winter (0.61 ± 0.11). FDiv in autumn (0.80 ± 0.09) and winter (0.79 ± 0.09) was significantly higher than it was in spring (0.76 ± 0.09) and summer (0.74 ± 0.09), and the latter two values differed significantly from each other.

*3.5. Influence of Land Cover Types on Avian Community Composition*

The bird community composition was significantly influenced by five of the eight environmental variables (seven land cover types and landscape diversity). Among the land cover types, woodland, built-up land, cultivated land, and water bodies showed the strongest correlations with variations in species composition across all seasons (Figure 3). Of these, woodland provided the highest explanatory rate (spring: 0.20; summer: 0.20; autumn: 0.24; winter: 0.12), followed by built-up land (spring: 0.15; summer: 0.16; autumn: 0.17; winter: 0.08) and cultivated land (spring: 0.11; summer: 0.11; autumn: 0.13; winter: 0.05). Additionally, the winter bird community was affected by bare land. In the ordination diagrams (Figure 3), we observed that the distribution of avian diet guilds was influenced by land cover type. For example, the PlantSeed guild (which feeds on plants and seeds) appeared more frequently in habitats with large areas of non-natural land cover, such as built-up and cultivated land, particularly in spring and autumn. During winter, the VertFishScav guild (which feeds on vertebrates, fish, and carrion) appeared more frequently in habitats with expansive visibility and a high proportion of water bodies. Insectivorous (birds that feed on invertebrates) and omnivorous birds were more evenly distributed in the ordination diagrams.

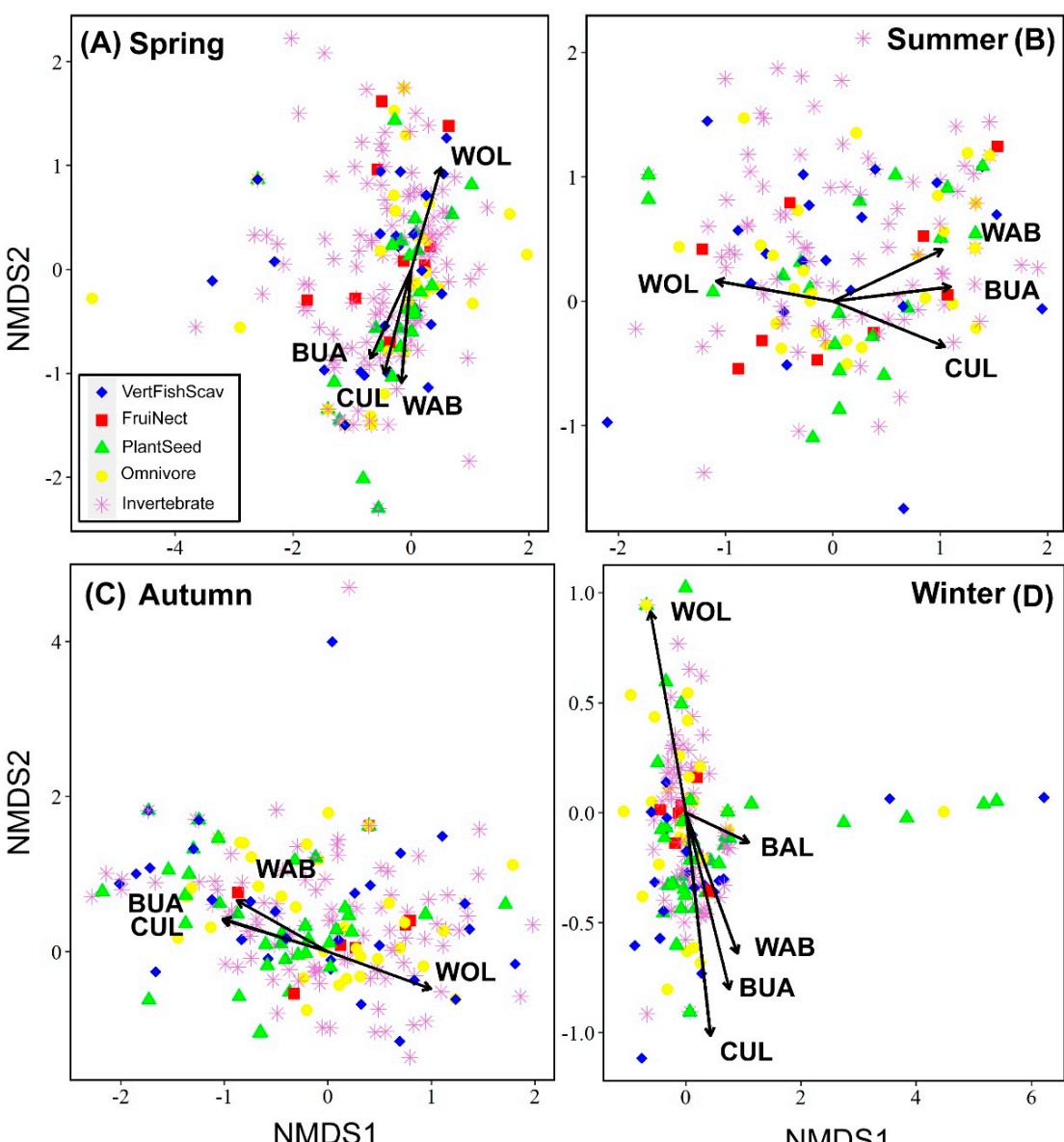

**Figure 3.** Non-metric multidimensional scaling (NMDS) showing the relationships between avian communities and land -cover types in the different seasons. Each symbol represents a species, and the shape of the symbol represents the feeding guild. Land cover types with significant effects are represented by arrows with labels. WOL = woodland; LD = landscape diversity; BUA = built-up areas; CUL = cultivated land; WAB = water bodies; BAL = bare land.

*3.6. Associations between Land Cover Types and Bird Functional Characteristics*

The fourth-corner analysis revealed numerous stable associations between land cover type and bird functional characteristics, particularly in the foraging strata (Figure 4). In most seasons, foraging in water was positively correlated with the proportion of water bodies and built-up land, whereas it was negatively correlated with the proportion of woodland. The opposite was true for foraging in the vegetation layer. We found that the functional characteristics of foraging in the vegetation layer were negatively correlated with the proportion of water bodies and non-natural land covers (i.e., built-up and cultivated land) but positively correlated with the proportion of woodland. Additionally, a positive relationship was observed between foraging on the ground and the proportion of cultivated land during spring and winter. In terms of diet, the relationships between most dietary characteristics and land cover type exhibited large seasonal variations, except for the

characteristic of the VertFishScav guild, which was significantly and positively associated with the proportion of water bodies. Body mass had a strong positive correlation with the proportion of built-up land and water bodies but showed a negative correlation with the proportion of woodland. This suggests that large-bodied birds occur more frequently in open habitats, whereas small-bodied birds are more common in dense habitats.

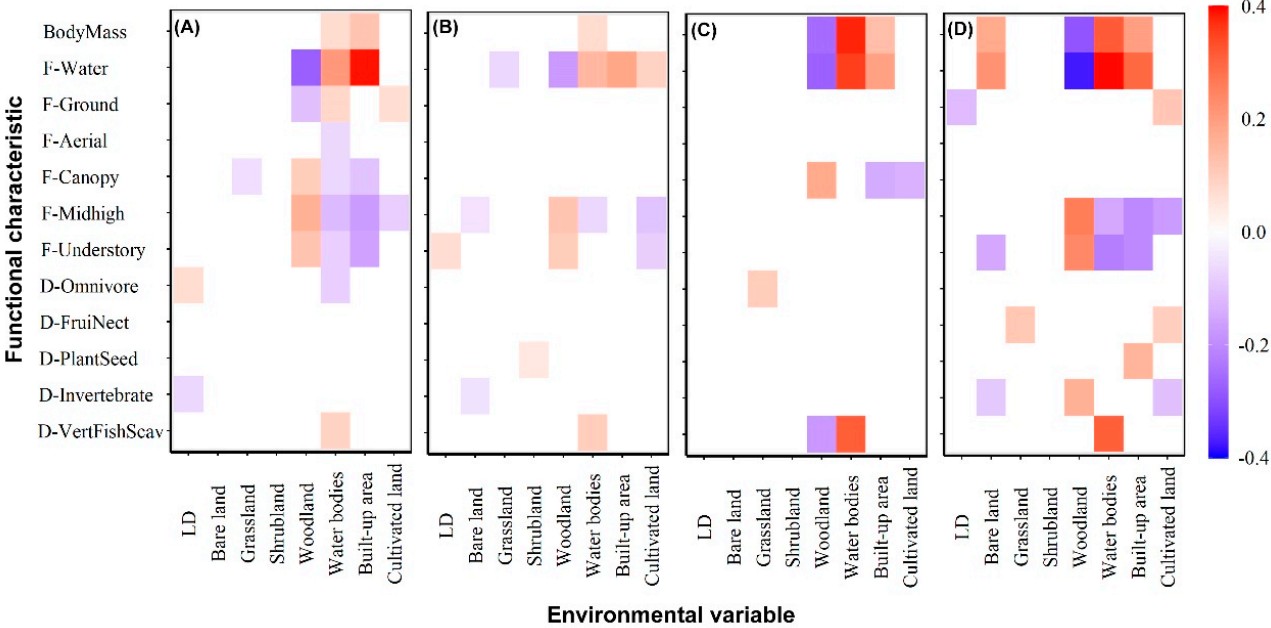

**Figure 4.** Interaction coefficient from fourth-corner analysis testing the relationship between bird functional characteristics and land cover in the different seasons: (**A**) spring; (**B**) summer; (**C**) autumn; (**D**) winter. Brighter squares show stronger associations than paler ones; positive associations are red, and negative associations are blue. F = foraging characteristics; D = diet characteristics.

## 4. Discussion

The preservation of biodiversity has emerged as a paramount concern in the context of the rapid decline in species worldwide. Montane environments are important biodiversity hotspots that have been subjected to habitat changes due to human activity. This study examined the impact of land cover types on avian communities in the mountainous region of Lishui, a typical urban area in Eastern China. Our results revealed that the diversity and composition of avian communities in mountainous urban areas exhibit seasonal variations and are influenced by land cover type. The results of this study provide a valuable resource for the future planning of urban areas in mountainous regions and the conservation of biodiversity.

Land cover distribution is closely associated with the survival of birds, making it a key factor influencing bird diversity [6,24]. Moreover, diverse land cover types offer varying resources and habitats for birds, potentially resulting in distinct impacts on avian diversity [8,33]. Forests or woodlands are typically the primary habitats for birds when they select nesting and roosting locations. This study identified woodland, cultivated land, and built-up land as the most dominant land cover types affecting avian diversity. The proportion of woodland had a negative effect on species richness and abundance, while that of non-natural land covers, such as built-up and cultivated land, had a positive effect. Although this finding is contrary to those of previous studies [8,34], these studies were primarily conducted in landscapes dominated by human activities, such as urban areas and farmland, where non-natural land cover predominated and woodland habitats were relatively scarce. In contrast, our research was conducted in a mountainous landscape characterized by a forest cover of over 80%, providing ample and abundant habitats for birds. Under these circumstances, as the proportion of woodland decreases and that of non-

natural landscapes increases, the variety of available habitats for birds also increases. In particular, generalist birds capable of inhabiting a variety of habitats with diverse resources can clearly derive benefits from the supplementary resources offered by non-natural land cover [9]. Similar results have been obtained in earlier studies conducted in mountainous regions [7,33]. However, it is important to note that some birds that specialize in specific habitats, especially those that are highly dependent on natural habitats, may not benefit from non-natural land cover [20]. This finding also reveals that the expansion of non-natural land cover could be strategically increased while preserving abundant natural forest or woodland remnants during mountain landscape development. Furthermore, we found that functional richness was negatively correlated with the proportion of woodland and positively correlated with non-natural land cover. This phenomenon can also be explained by the habitat diversity hypothesis, which states that niche diversity increases with habitat type [15]. Mosaic forest landscapes, characterized by anthropogenic land cover interspersed with woodland areas, offer a greater number of ecological niches than more homogenous landscapes do [7,35]. Consequently, they support higher functional richness than do areas that are entirely forested.

In this study, bird community diversity also exhibited seasonal variations, with higher richness and abundance in spring and autumn and lower richness and abundance in winter (Figure 2). The availability of food resources, including variety, distribution, and quantity, is considered one of the most influential factors [14,36]. In contrast to the scarcity of food resources during the colder winter months, there is a notable increase in both the variety and quantity of food available during spring and autumn. For example, flowering plants can offer ample pollen and nectar resources, whereas cultivated land provides a substantial quantity of invertebrates and seed grains following spring plowing. Autumn is the period during which various types of grains and fruits mature. In addition, the reasons for the low number of birds in winter may include bird migration, hibernation, and decreased activity aimed at minimizing heat expenditure. Species richness and functional richness cannot be described by simple linear relationships [37]. Theoretically, an increase in species richness typically leads to an increase in functional richness. However, functional redundancy may occur when additional species fill the same ecological role [38,39]. In this study, the number of species observed in autumn exceeded that in spring and summer, whereas functional diversity exhibited a notable decrease (Figure 2). Functional evenness describes the regularity of the distribution of functional traits within a functional space and is related to resource utilization [40]. The greater the value, the more comprehensive the utilization of resources. We found that the bird community exhibited the highest utilization of environmental resources during summer, followed spring and autumn, with the lowest utilization during winter (Figure 2). Functional divergence describes the differences in functional characteristics within a community and is related to niche differentiation and resource competition [40]. We found that resource competition within the bird community was less pronounced during autumn and winter but more intense during spring and summer (Figure 2).

Species distribution in avian communities is influenced by a combination of environmental conditions and species traits [6,8,17]. Birds with narrow dietary niches are restricted to a limited number of locations where their specific niche requirements are met. For example, in the ordination diagrams, the guild of VertFishScav tended to inhabit areas in close proximity to water bodies, especially in winter, whereas those guilds that feed on plants and seeds were more commonly observed in habitats characterized by a high proportion of built-up and cultivated land, typically in the vicinity of home gardens and farmland (Figure 3). On the other hand, birds with wide niches thrive in various habitats [13]. For example, the omnivorous birds in our study are evenly distributed in the ordination diagrams (Figure 3). Moreover, environmental conditions within a given habitat fluctuate with the changing seasons, and the reliance of birds on specific habitats fluctuates throughout the year (Figures 3 and 4). Consistent with previous findings [41,42], our study also revealed a close association between land cover type and the foraging strata

of avian species. These findings have important implications for the construction of cities in mountainous regions. In the urban planning and construction process, it is imperative for planners to prioritize landscape configurations, particularly the type of land cover, to create a conducive environment for diverse bird species that occupy different ecological niches. Moreover, our findings indicate a significant positive correlation between body mass and the proportion of built-up land and water bodies, and a negative correlation with the proportion of woodland (Figure 4). This suggests that larger-bodied birds are more likely to occur in open habitats, whereas smaller-bodied birds are more prevalent in dense habitats. Smaller body sizes result in more rapid heat consumption [43]. To offset energy expenditure, smaller birds typically need to forage more frequently, therefore possibly attracting more attention from predators and humans. Consequently, they inhabit enclosed environments with dense vegetation to minimize exposure risks. In terms of key species, we found that among the 186 surveyed line transects, 159 contain China's key protected birds, and 153 transects have species listed as threatened on the Red List. This once again demonstrates the importance of mountainous areas in conserving biodiversity. Changes in landscape patterns in mountainous areas could affect these rare and endangered species. In future study, we will focus on this aspect.

## 5. Conclusions

Our results indicated that mountainous landscape patterns have considerable effects on avian diversity and community structure. Non-natural land cover can increase the taxonomic diversity and functional richness to a certain extent. However, this increase was observed when an adequate natural environment was maintained. Hence, it is imperative that we continue to protect the natural environment and enact measures to prevent habitat destruction. Furthermore, we found that various land cover types possess distinct resource endowments that influence bird distribution. Therefore, we recommend that in the planning of urban mountain landscapes, attention should be paid not only to the extent of natural landscape development but also to the functions provided to organisms by different land cover types. Maintaining diverse landscapes enables different species to occupy different niches and survive in mountainous urban areas, thereby preserving biodiversity.

**Supplementary Materials:** The following supporting information can be downloaded at: https://www.mdpi.com/article/10.3390/d16020107/s1: Table S1. Bird species recorded in the study area.

**Author Contributions:** Conceptualization, C.L.; methodology, W.Z. and S.Z.; software, W.Z. and S.Z.; formal analysis; W.Z. and S.Z.; investigation, X.Y., J.T., D.C., Y.Y. and J.S.; writing—original draft preparation; X.W.; writing—review and editing, P.C. All authors have read and agreed to the published version of the manuscript.

**Funding:** This research was funded by the Open Research Fund of the State Environmental Protection Key Laboratory on Biodiversity and Biosafety (202305210623-4) and the Jiangsu Academy of Forestry Youth Foundation (JAF-2022-01).

**Data Availability Statement:** The data presented in this study are available on request from the corresponding author. The data are not publicly available due to data protection restrictions.

**Acknowledgments:** We would like to thank the Lishui Municipal Bureau of Ecological Environment for their support of this study and the people who participated in the bird field survey.

**Conflicts of Interest:** The authors declare no conflicts of interest.

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
