# Peer review of "Effects of Land Cover on the Taxonomic and Functional Diversity of the Bird Communities on an Urban Subtropical Mountain"

_diversity, doi:10.3390/d16020107_

Round 1

Reviewer 1 Report

Comments and Suggestions for Authors

In publication Rahbek et al. 2019 about 25% of all land area, mountain regions are habitats to more than 85% of the world's species of amphibians, birds, and mammals. The concept of bird studies in a mountain city is an important aspect of the ecology of these animals. At the same time, they are very important due to their protection in mountain ecosystems.

l. 43: publication [2] does not refer to vertebrates.

l. 43: publication [3] does not refer to reptiles.

l. 93-97: no research hypothesis for the results: [l. 221] section - .3. Influence of season on avian community diversity.

l. 115: Each transect was surveyed once per season. Explain what the season means?

l. 117-118: please provide references of the method used. How many counts were carried out in the morning and how many in the evening? How long did the counting take in one day. How many people/teams counted birds. How long did the inspection of one transect take?

l. 216-220: table 3. - autumn data not included.

l. 221: 3.4. Influence of season on avian community diversity. There is no research hypothesis and no research purpose for these results.

l. 307: references no. [8] – refers to the city; reference no. [31] - refers to the agricultural landscape, therefore these results are not comparable. Analyze again. There are many works on birds in the mountains and in cities located in the mountains, e.g.:

Jankowski, J. E., Ciecka, A. L., Meyer, N. Y., & Rabenold, K. N. (2009). Beta diversity along environmental gradients: implications of habitat specialization in tropical montane landscapes. Journal of Animal ecology, 78(2), 315-327.

Jankowski, J. E., Merkord, C. L., Rios, W. F., Cabrera, K. G., Revilla, N. S., & Silman, M. R. (2013). The relationship of tropical bird communities to tree species composition and vegetation structure along an Andean elevational gradient. Journal of Biogeography, 40(5), 950-962.

Dehling, D. M., Fritz, S. A., Töpfer, T., Päckert, M., Estler, P., BöhningGaese, K., & Schleuning, M. (2014). Functional and phylogenetic diversity and assemblage structure of frugivorous birds along an elevational gradient in the tropical Andes. Ecography, 37(11), 1047-1055.

Xu, W., Zheng, D., Huang, P., Yu, J., Chen, Z., Zhu, Z., ... & Fu, W. (2022). Does Bird Diversity Affect Public Mental Health in Urban Mountain Parks?—A Case Study in Fuzhou City, China. International Journal of Environmental Research and Public Health, 19(12), 7029.

Pineda-López, R., Tepos-Ramírez, M., Acosta-Ramírez, A., Calderón, A. M. S., & Feregrino, A. O. (2023). Elevational and seasonal changes in a bird assemblage within a mountain system in central Mexico. Ornithology Research, 1-8.

l. 317: references nr [31]. This finding also reveals that the expansion of non-natural .

l. 323-326: provide a citation to support this statement.

l. 327-351: part not supported by the research hypothesis.

l. 364-377: these are the research results described in the discussion section.

l. 289-382: in the discussion chapter, a lot of my own results are described. Discuss results with more literature.

Author Response

/

Reviewer 2 Report

Comments and Suggestions for Authors

The authors write in the introduction that mountain areas maintain high biodiversity, but the study makes no reference whatsoever to changes in this diversity in relation to the urbanization gradient. Trivial research hypotheses are posed, from which little is derived. The analyses carried out, which are not always methodologically correct, do not contribute anything new to science. The discussion is very general, does not further analyze the changes taking place and smoothly powadz to the conclusion that forest fragmentation has a positive effect on birds. 

The methodology should be improved and expanded so that, based on the data collected, information on the actual impact of habitat changes on biodiversity can be obtained, taking into account specialized, rare and endangered species, rather than general species abundances.  

Comments on the Quality of English Language

No comments.

Author Response

/

Round 2

Reviewer 1 Report

Comments and Suggestions for Authors

l. 65. publication no. [17] – please add: Tryjanowski, P., Morelli, F., Mikula, P., Krištín, A., Indykiewicz, P., Grzywaczewski, G., & Jerzak, L. (2017). Bird diversity in urban green space: A large-scale analysis of differences between parks and cemeteries in Central Europe. Urban Forestry & Urban Greening, 27, 264-271.

l. 77-78: incorrect quotation.

l. 226-229: table 3 - data from autumn are still not included.

l. 300-308 – no publication citation.

Discussion - it is worth discussing the occurrence of ecological groups of birds (insectivorous, predatory, herbivorous, fruit-eating, etc.) divided into seasons.

Please attach a list of bird species found during the survey, divided into breeding and non-breeding species, divided into spring-summer-autumn-winter.

Author Response

/

Reviewer 2 Report

Comments and Suggestions for Authors

Dear Authors, 

the corrections and clarifications are suitable. 

Comments on the Quality of English Language

English is correct.

Author Response

/
